# Chronic Kidney Disease-Associated Immune Dysfunctions: Impact of Protein-Bound Uremic Retention Solutes on Immune Cells

**DOI:** 10.3390/toxins12050300

**Published:** 2020-05-06

**Authors:** Maxime Espi, Laetitia Koppe, Denis Fouque, Olivier Thaunat

**Affiliations:** 1Service de Transplantation, Néphrologie et Immunologie Clinique, Hôpital Edouard Herriot, Hospices Civils de Lyon, 69000 Lyon, France; maxime.espi@inserm.fr; 2CIRI, INSERM U1111, Université Claude Bernard Lyon I, CNRS UMR5308, Ecole Normale Supérieure de Lyon, 69000 Lyon, France; 3Département de Néphrologie-Dialyse-Nutrition, Centre Hôpital Lyon Sud, Hospices Civils de Lyon, 69310 Pierre Bénite, France; laetitia.koppe@chu-lyon.fr (L.K.); denis.fouque@chu-lyon.fr (D.F.); 4CarMeN, INSERM U1060, INRA 1397, 69310 Pierre-Bénite, France; 5Lyon-Sud Medical Faculty, Université de Lyon, 69000 Lyon, France; 6Lyon-Est Medical Faculty, Université de Lyon, 69000 Lyon, France

**Keywords:** chronic kidney disease, uremic toxins, immune system

## Abstract

Regardless of the primary disease responsible for kidney failure, patients suffering from chronic kidney disease (CKD) have in common multiple impairments of both the innate and adaptive immune systems, the pathophysiology of which has long remained enigmatic. CKD-associated immune dysfunction includes chronic low-grade activation of monocytes and neutrophils, which induces endothelial damage and increases cardiovascular risk. Although innate immune effectors are activated during CKD, their anti-bacterial capacity is impaired, leading to increased susceptibility to extracellular bacterial infections. Finally, CKD patients are also characterized by profound alterations of cellular and humoral adaptive immune responses, which account for an increased risk for malignancies and viral infections. This review summarizes the recent emerging data that link the pathophysiology of CKD-associated immune dysfunctions with the accumulation of microbiota-derived metabolites, including indoxyl sulfate and p-cresyl sulfate, the two best characterized protein-bound uremic retention solutes.

## 1. Introduction

Chronic kidney disease (CKD) is defined by abnormalities of kidney function or structure present for more than 3 months. The definition of CKD includes all individuals with markers of kidney damage or those with an estimated glomerular filtration rate (eGFR) of less than 60 mL/min/1.73 m^2^ on at least two occasions 90 days apart [1]. CKD severity is classified into five stages according to the level of GFR. Patients at the last (end) stage of CKD require renal replacement therapy: either chronic hemodialysis or renal transplantation.

Affecting more than 10% of the global population in developed countries, CKD has changed from a subspecialty issue to a global health concern. All stages of CKD are indeed associated with increased risks of premature mortality and/or decreased quality of life, inducing a high economic cost to health systems [2].

Regardless of the primary disease responsible for the destruction of kidneys, patients suffering from CKD are characterized by a similar comorbidity profile, which includes: mineral-bone disorders, anemia, insulin-resistance, sarcopenia, and cognitive impairment [1]. Progress over the last decades has linked these alterations with the accumulation of uremic toxins, i.e., solutes normally excreted by the kidneys, that impair normal cell physiology [3]. In addition to these observations, CKD patients are characterized by a complex impairment of the immune system, which combines low-grade chronic inflammation and the inability to mount protective immune responses. CKD-induced chronic inflammation is associated with a marked increase in cardiovascular mortality [4], while the inability to mount an efficient immune response increases the risk of malignancy [5] and infection [6,7,8]. The recent pandemic of SARS-Cov-2 provided a vivid illustration of the latter problem: COVID-19 CKD patients had more pronounced lymphopenia, developed less anti-SARS-Cov-2 effectors, and had an increased risk of death as compared to non-CKD patients [9]. 

The pathophysiological mechanisms underlying CKD-associated immune dysfunctions have long remained unclear but recent evidence suggests that the accumulation of certain uremic toxins is responsible for the dysfunction of immune cells. 

## 2. Protein-Bound Uremic Retention Solutes

### 2.1. Classification of Uremic Retention Solutes (URS)

The current classification for uremic retention solutes [10] distinguishes three subtypes based on their size: free-water soluble low molecular weight molecules (<500 kDa), middle molecules (500–60,000 kDa), and protein-bound uremic retention solutes (Table 1). 

Free-water soluble low molecules and middle molecules are mainly endogenous molecules, participating in normal biologic functions. For example, urea and phosphate are considered as free-water soluble uremic toxins, because of their known toxicity on glucose homeostasis [11] and on cardiovascular system [12]. Likewise, parathyroid hormone and β2-microglobulin are two middle molecules with essential biological functions, the toxicity of which is due to their accumulation during CKD [3]. 

In contrast, the third class of uremic retention solutes essentially gathers exogenous molecules, which derive from dietary intake, and bind to proteins (hence their name: protein-bound uremic retention solutes, PBURS). Only a small portion of PBURS (approximately 5% [13]) circulate as a free fraction and may exert a biological effect [14]. Many observational studies have demonstrated the association between serum levels of PBURS and mortality/complications in CKD [15].

### 2.2. CKD-Associated Dysbiosis and PBURS

The gastrointestinal tract (GIT) is inhabited by approximately 100,000 billion bacteria and other microorganisms, collectively known as the “gut microbiota”. The alteration in the composition and function of gut microbiota is a condition called dysbiosis. The causal roles of dysbiosis in host pathophysiology like obesity, diabetes mellitus, non-alcoholic fatty-liver disease, malignancies, and major depressive disorders have been increasingly recognized [16].

During CKD, numerous external and internal factors associate to promote bacterial overgrowth in the small intestine and alterations in microbiota composition [17]. This “uremic dysbiosis” leads to an increased production of PBURS [18], which accumulate in the internal milieu of CKD patients due to an increased permeability of the gut epithelial barrier [19] and a decrease of renal elimination [13]. The role of the microbiota in PBURS overload during CKD has been demonstrated by the fact that both CKD-germ free mice [20] and CKD patients without colon [21] have PBURS levels close to zero. 

The link between PBURS and CKD-associated immune dysfunction has recently emerged. Although many different PBURS have been identified (Table 1) [3,13], this review will focus on the indoles, i.e., indoxyl sulfate (IS) and phenols (i.e., p-cresyl sulfate (pCS)), which are tryptophan and tyrosine catabolites, respectively, the effects of which on the immune system are currently the best characterized.

### 2.3. Tryptophan Catabolites

Tryptophan is an essential amino-acid which cannot be synthetized by human metabolism and must therefore be provided by the diet [22]. Tryptophan is catabolized in the colon by numerous gut bacterial species [22,23,24] into indole (through a tryptophanase enzyme), but also a variety of other derivatives. In physiological conditions, tryptophan catabolites are beneficial for health, and are known to enhance intestinal barrier function [25,26], to promote anti-inflammatory [27,28] and anti-oxidative responses [29], and even to extend health span in different animal models [30]. 

Systemic effects of tryptophan catabolites were the subject of numerous studies in recent years, and it is now widely accepted that they mediate their effect through the aryl hydrocarbon receptor (AhR) pathway [22,31]. The AhR is a member of the Per-ARNT-Sim-basic helix-loop-helix protein family which acts as a transcription factor following ligands binding [32]. AhR is normally kept inactive in the cytosol by a chaperone complex. Upon agonist binding, the AhR complex translocates into the nucleus where it binds DNA-responsive elements to regulate gene expression [32].

During CKD, regulation of tryptophan catabolism is disrupted (Figure 1). CKD patients exhibit significant expansion of bacterial families possessing tryptophanase (including *Bacteroides*). This dysbiosis [18,33] is responsible for an increased production of indoxyl sulfate (IS, transformed from indole by liver enzymes cytochrome P450 2E1 and sulfotransferase [22]), which is also less secreted by tubular epithelial cells. This results in a two to 40-fold increase in free IS concentration in CKD patients stage three to five [13]. 

### 2.4. Phenols Derivates

Phenols are derived from tyrosine and phenylalanine metabolism. Theses aromatic amino acids are converted into phenolic compounds (such as phenol and p-cresol) through a series of transamination, deamination, and decarboxylation reactions by bacterial fermentation in the distal part of the colon. Detoxification of phenols occurs in the mucosa of the colon and in the liver, where p-cresol is sulfated into p-cresyl sulfate (pCS) by an aryl-sulfotransferase. 

Preliminary studies of the functional characteristics of intestinal microbiota of CKD patients have reported a higher prevalence of phenol-producing bacteria (*Enterobacteriaceae* and *Enterococcaceae* families) and of p-cresol-producing bacteria (including *Clostridium perfringens*). As for tryptophan catabolites, accumulation of pCS during CKD results not only from the dysbiosis [34] but also from decreased renal elimination [35]. The free fraction of pCS is two to five fold higher in CKD patients stage three to five [13]. 

In contrast with IS, the biological effects of pCS cannot be explained by a unique molecular pathway. For instance, pCS impairs insulin signaling through activation of kinases (ERK1/2) in muscular skeletal cells [36], while in tubular epithelial cells [37,38] and leukocytes [39] pCS activates the nicotinamide adenine dinucleotide phosphate (NADPH) oxidase, leading to local reactive oxygen species (ROS) production.

### 2.5. Role of PBURS in CKD-Associated Complications

Over the past decades, clinical and experimental data have linked the accumulation of PBURS, and most notably pCS and IS, with the pathophysiology of CKD-associated complications [15,40], including vascular calcification and atherosclerosis [41,42,43], anemia [44], insulin-resistance [36], and mineral-bone disorders [45]. Whether IS and pCS are also involved in CKD-associated immune dysfunctions is less well established, but accumulating evidence, systematically reviewed below, supports this theory.

Of note, although the highest serum PBURS levels are found in patients on hemodialysis, the present review will focus on studies dealing with previous stages of CKD. Indeed, it is well known that hemodialysis therapy itself, through the mechanical stimulation from blood pumping, the pollution of dialysate by bacterial metabolites [such as endotoxin or lipopolysaccharide (LPS)), and the limited biocompatibility of the dialysis membrane, can perturbate the immune system independently from PBURS [46].

## 3. CKD Induces Chronic Activation of Innate Effectors and Endothelial Damages 

The innate immune system, which forms the most ancestral and first line of defense of the organism, consists of different cell subsets and the soluble proteins of the complement system. The innate immune system provides an immediate (although non-specific) response to pathogens [47]. Innate effectors, including neutrophils and monocytes, are classically activated via pattern recognition receptors (including toll-like receptors, TLRs), which bind molecular motives broadly shared by pathogens and endogenous molecules released following cell stress/injury. Activation of innate immune cells leads to the release of pro-inflammatory cytokines, which have an important protective role but can turn deleterious in cases of chronic exposure [48].

CKD is characterized by a remarkable increase in pro-inflammatory cytokine levels, in particular tumor necrosis factor α (TNFα) and interleukin 6 (IL-6) [49,50], which are inversely correlated with the decline in GFR [51,52]. Neutrophils and monocytes from CKD patients display an exaggerated response to stimulation with lipopolysaccharide (LPS) [53], which could be due to the fact that the uremic environment induces an increased expression of TLR2 and 4 [49,54,55]. The permanent activation of monocytes during CKD could also be the direct consequence of PBURS. IS indeed activates the aryl hydrocarbon receptor [56] of monocytes, thereby promoting the generation of pro-inflammatory cytokines [57], which in turn provokes endothelial damage [58] and increases the risk of cardiovascular complications [59].

This problem is worsened by an amplification loop due to the direct effects of PBURS on endothelial cells. Several studies have indeed documented that pCS impairs vascular remodeling and induce vascular contraction [60], while IS alters endothelial cell functions [61,62,63]. In particular, the uremic milieu alters the ability of endothelial cells to control the alternative complement pathways, which in turn amplifies the endothelial injuries [64]. In line with these data, decreasing IS rate with adsorbent carbon particles has been proven efficient to reduce the levels of endothelial dysfunction markers [65].

Finally, the demonstration of the involvement of PBURS in triggering the deleterious crosstalk between endothelial cells and innate immune effectors during CKD has been provided by experimental data showing that the mere administration of IS or pCS in rats induced leukocytes adhesion and extravasation, leading to interrupted blood flow [66]. 

## 4. Neutrophils Responses Against Extracellular Bacteria are Impaired During CKD

Constant activation of the innate immune effectors during CKD does not imply that patients are better protected against pathogens. Instead, CKD is characterized by a dramatic increase in the incidence and the severity of infectious episodes [7]; the risk of death from infectious disease of CKD patients is indeed estimated 10 to 1000 fold higher than that of healthy age-matched controls [6,8].

The first line of defense against extracellular bacteria is provided by neutrophils, which are the most abundant type of white blood cells in mammals. Neutrophils provide protection by migrating fast (within minutes) to the site of infection, and by attacking micro-organisms through phagocytosis (ingestion and destruction of bacteria). Although uremic milieu has been shown to trigger apoptosis of neutrophils in culture [67], there is no clear evidence that the number of neutrophils is reduced in CKD patients [68,69]. In contrast, there is a wide consensus that the neutrophils of CKD patients display defective functions. First, some reports indicate that certain uremic toxins can inhibit neutrophils chemotaxis [70,71]. Second, there is a wide consensus that phagocytic functions are defective in the neutrophils of CKD patients [67,72]. The fact that: (i) in vitro phagocytic capacities of neutrophils from CKD patients were improved when the cells were cultured in a non-uremic medium, and that (ii) conversely, the phagocytic capacities of neutrophils from healthy patients were altered when the cells were cultured in an uremic milieu, suggests that uremic toxins are directly responsible for the problem (Table 2). 

A central role in the destruction of ingested bacteria by neutrophils is played by the nicotinamide adenine dinucleotide phosphate (NADPH) oxidase, an enzyme that converts oxygen to superoxide free radicals. Many uremic retention solutes are able to inhibit NADPH oxidase activity [73,74], including IS [73] and pCS [73,75]. The molecular mechanism by which these uremic toxins impair the NADPH oxidase activity is not entirely clear but seems to involve an inhibition of neutrophils metabolism, leading to a state of depressed cell energy production [74,76,77]. 

If the defects in neutrophils functions provide a likely explanation for the increased risk of bacterial infectious complications observed in CKD patients, this mere mechanism cannot account on its own for many other typical features of CKD-associated immune dysfunctions. 

## 5. Adaptive T-Cell Responses are Impaired in CKD Patients

The first evidence that CKD is associated with a defect in adaptive T cell responses came from the observation made in the mid-1950s’ that survival of skin homografts is prolonged in uremic patients [99] (rejection of skin graft, is indeed strictly dependent on T cell-mediated rejection [100]). Given the critical role of T cells both in cancer immunosurveillance [101] and the elimination of intra-cellular pathogens (in particular viruses), CKD-induced defects in adaptive T cell responses are likely responsible for the increased risk for malignancies [5,102,103] and severe viral infections (including COVID-19 [9]) observed in uremic patients.

Jawed vertebrates, have sophisticated adaptive immunity that can mount two types of specific effector responses following exposure to an antigen: humoral (i.e., consisting of antibodies), or cellular (depending on CD8+ cytotoxic T cells) [47]. Both types of adaptive responses require the activation of T cells by antigen presenting cells (APC), the main type of which is dendritic cells (DCs), to be initiated [47]. An important feature of adaptive immunity is the generation of memory cells, which respond more rapidly and more efficiently upon exposure to the same antigen [47].

DCs are found in reduced number in the circulation of CKD patients [80,81] and this decrease has been shown to parallel the decline in GFR [82]. Furthermore, DCs from CKD patients express less major histocompatibility complex (MHC) class I and class II, and costimulatory molecules both at baseline [83], and following in vitro stimulation [104]. As expected from these phenotypic abnormalities, DCs from CKD patients showed reduced capacity to activate T cells in vitro [85]. The fact that: (i) DCs from CKD patients exposed to non-uremic milieu partially recover a normal phenotype [84], and (ii) conversely DCs from healthy controls cultured in uremic milieu display a decreased expression of costimulatory molecules, suggests an important role for uremic toxins (Table 2). In line with this proposal, high pCS concentrations induce remarkable alteration of DCs functions, including reduced phagocytosis and antigen processing and presentation [78,79] (Table 2). In vitro IS exposure also has an impact on DCs, leading to a decrease in proliferation and expression of costimulatory molecules [86], likely through activation of AhR [87] (Table 2). In line with this theory is the fact that synthetic agonists of AhR have been shown to reduce DC proliferation and to promote the acquisition of a tolerogenic phenotype [105] in vitro, which translates into decreased allergic lung inflammation [106] and the prevention of rejection of pancreatic islet allografts [107] in murine models. 

There are conflicting data regarding whether or not CKD induces a (moderate) reduction in the number of circulating T cells [68,88,108]. In contrast, it is widely accepted that CKD has a profound impact on T cell subset distribution. CKD patients are indeed characterized by a decreased number of naïve T cells [89] due to both reduced thymic output [90] and increased apoptosis [88]. In contrast, effector memory T cells remain in normal numbers in CKD patients [89], who are also characterized by an expansion in terminally differentiated CD4+ CD28^null^ T cells [92,93] (Table 2). These abnormalities, together with the observation that CKD patients have a less diverse T-cell receptor (TCR) repertoire than healthy controls [94], are evocative of a compensation by an increased peripheral proliferation of memory T cells. This hypothesis has been confirmed by the demonstration that peripheral T cells of CKD patients have reduced relative telomere length [90]. Altogether these data demonstrate that CKD is associated with premature ageing of the T cell compartment [68,91], a process that may be induced/accelerated by uremic milieu-induced chronic-low grade inflammation (see paragraph 3 above). 

Several lines of experimental evidence directly link accumulation of PBURS with defective T cell responses. First, increasing the rate of pCS in mice correlates with decreased production of IFNγ by T helper type 1 (Th1) cells [95]. Furthermore, CD4+ T cells are particularly sensitive targets for AhR regulation. Activation of AhR by exogenous ligands promotes the development of regulatory T cells while suppressing effector T cell responses both in vitro [109] and in various murine transplantation models, including graft-versus-host disease [110], cardiac allograft [111], and pancreatic islet allograft [107]. Furthermore, treatment of mice with AhR ligands has also been shown to ameliorate the development of several T-cell dependent auto-immune diseases [112,113]. 

## 6. CKD Induces Defective Humoral Responses

In addition to their inability to mount efficient T cell responses, CKD patients also exhibit impaired adaptive humoral responses. For instance, the proportion of patients with end stage kidney disease able to develop protective titers of anti-HBs antibodies following vaccination was only 60% in the 1980s’ (versus 95% for healthy controls) [114]. Despite intensification of the administration of the vaccine in the new protocols, the rate of responders is still stagnating below 80% today [115,116]. 

Generation of antibodies against a protein antigen starts, within secondary lymphoid organs, with the binding of the antigen to the surface B cell receptor (BCR). This delivers the first signal of activation to cognate B cells and promotes the internalization of the antigen, which is then processed and presented on the cell surface within MHC molecules. The second signal of activation is delivered by a specific subset of CD4+ T cells, named T follicular helper (Tfh) cells. The latter recognize the antigen/MHC complexes on B cell surface through their TCR and in return deliver soluble (interleukin 21) and membrane-bound (CD40 ligand, inducible costimulator ICOS, etc.) costimulatory signals. Activated B cells then enter the germinal center reaction leading to the generation of high affinity memory B cells and antibody-producing plasma cells [47]. 

From the paragraph above, it is intuitive that the alterations of helper T cell responses induced by uremic milieu can indirectly explain the defect in antibody responses observed in CKD patients. In line with this hypothesis, a study has reported that patients on hemodialysis generate less antigen-specific memory CD4+ T cells after vaccination with HBV antigen [117]. Regarding the impact of uremic milieu on the specific Tfh subset, there is scarce, conflicting evidence [115,118] that CKD could be associated with a reduction in their number (in the circulation), however, data that would document the impact on Tfh functions are currently lacking. 

Another (non-exclusive) explanation for the defect in antibody response of CKD patients is the direct impact of uremic milieu on B cells. Decrease in GFR correlates with a decrease in total B cell number [68,96], a problem that seems to affect both naïve (CD19+ CD27−) and memory (CD19+ CD27+) populations [97] and that could be related to a reduced expression in receptor for the survival factor BAFF (for B-cell activating factor) [97]. In line with this theory, it has been shown that B cells from CKD patients have an increased susceptibility to apoptosis in vitro [96], which was associated with a decreased expression of B-cell lymphoma 2 gene (Bcl-2) [96], a prosurvival factor induced by BAFF [119]. 

PBURS appear as major factors for CKD-induced B cell alterations. In a mice model, chronic administration of pCS induced a strong decrease in B cell count [98]. Additionally, although the effect of uremic levels of tryptophan-derived URS (including IS) on B cells has not been studied yet, it is worth mentioning that a growing number of studies suggests that the AhR pathway is an important regulator of several aspect of B cell biology. First, B cells express AhR and its activation inhibits B cell lymphopoiesis [120]. Second, it has long been observed that mice treated with AhR agonists are unable to mount a humoral response following immunization [121]. B-cell proliferation following BCR stimulation requires AhR [122], which serves a critical role in regulating activation-induced cell fate outcomes. AhR indeed negatively regulates class-switch recombination, represses differentiation of B cells into antibody-secreting plasma cells [123], and promotes the acquisition of a regulatory and anti-inflammatory profile (IL-10 producing Breg) [124]. 

## 7. Conclusions and Perspectives

CKD is associated with a complex defect of almost all components of the immune system (summarized in Figure 2 and Table 2), resulting in a peculiar state of “chronic inflammatory immune depression” named CKD-associated immune dysfunctions. It is now widely accepted that the accumulation of PBURS is a critical factor explaining the dysfunction of innate immunity (Table 2), which accounts for both the increased risk for bacterial infections and the damage to endothelial cells. However, CKD-associated immune dysfunctions also include impaired adaptive cellular and humoral responses, which are critical for the protection against intra-cellular pathogens and cancer. Indirect evidence also suggests that PBURS could also be responsible for these defects (Table 2) but additional mechanistic studies are required to formally demonstrate this link.

Admitting that PBURS indeed represent the common pathophysiologic factor triggering the various CKD-associated immune dysfunctions, how can we take advantage of this information to improve the immune status of patients? In this regard, hemodialysis appears to be of little help. First it cannot be used in early stages of CKD. Second, even at a later stage, PBURS are not removed efficiently with this technique because most of the toxins are, by definition, bound to protein (while only the free fraction can diffuse across the dialysis membrane). Since PBURS are derived from diet protein, adaptation of intakes (i.e., reduction of tryptophan) could be beneficial by restoring the catabolites balance. Given the importance of CKD-associated dysbiosis in PBURS generation, it is also tempting to target the gut microbiota through pre/probiotics [18] or tryptophanase inhibitors [125]. Preventing PBURS absorption by chelating the toxins in the intestinal tract has also been proposed, with AST-120 for instance [126]. Finally, another theoretical therapeutic possibility would be to act later in the cascade by blocking the AhR pathway, which seems pivotal to mediate IS effects on the immune effectors. Beyond their interest as therapeutic targets, PBURS could also be interesting biomarkers to stratify the risk of immune-mediated complications in CKD. Until recently, the quantification of PBURS was not available in routine clinical practice due to its cost and the need of specific technologies (such as mass spectrometry [127]). However, the recent development of cheap assays that allow measuring non-invasively the levels of PBURS in the saliva, has the potential to rapidly change this situation [128,129]).

## Figures and Tables

**Figure 1 toxins-12-00300-f001:**
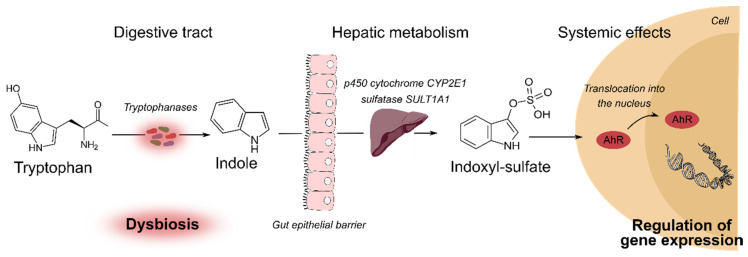
Generation and impact of indoxyl sulfate (IS). During chronic kidney disease, dysbiosis increases tryptophanase-producing bacterial species that convert tryptophan into indol. IS is derived from indole hepatic metabolism. The loss of renal function lead to decreased excretion of IS. IS acts as AhR ligands, permitting its translocation into the nucleus of various cells, where it controls the expression of various genes. Abbreviations are; IS: indoxyl sulfate; AhR: aryl hydrocarbon receptor.

**Figure 2 toxins-12-00300-f002:**
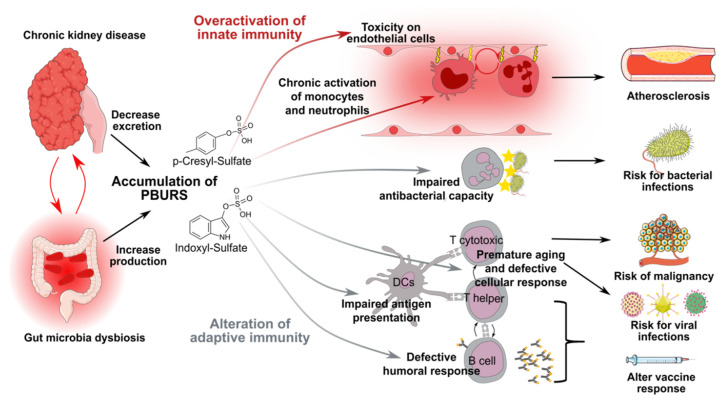
Schematic representation of chronic kidney disease (CKD)-associated immune dysfunctions. Chronic kidney disease and gut-microbiota dysbiosis lead to the accumulation of protein-bound uremic retention solutes (PBURS), including p-cresyl sulfate and indoxyl sulfate, which have an impact on innate and adaptive immune systems. PBURS impair endothelial cells function and induce chronic low-grade activation of innate immune effectors (monocytes and neutrophils). This toxic loop is responsible for accelerated atherosclerosis. Despite chronic activation, the antibacterial capacity of neutrophils is impaired by PBURS. PBURS also affect the adaptive immune system. CKD patients are characterized by defective dendritic cells (DCs), premature aging of T cells and impaired cellular and humoral responses, which in turn account for an increased risk for malignancies and viral infections. Abbreviations: DCs: dendritic cells; PBURS: protein-bound uremic retention solutes.

**Table 1 toxins-12-00300-t001:** Current classification of major uremic retention solutes.

	Low Molecular Weight Molecules (<500 kDa)	Middle Molecules (500–60,000 kDa)	Protein-Bound Uremic Retention Solutes
**Selection of clinically relevant molecules**	-Urea -Phosphate -Uric acid -Creatinine -Carbamylated compounds -Trimethylamine-N-oxide *	-B2 microglobuline -Parathyroid hormone -Fibroblast-growth-factor 23 -Atrial natriuretic peptide -Interleukin 6, 8, 10 -TNFα	-Indoxyl sulfate * -P-cresyl sulfate * -Indole-3- acetic acid * -Kynurenic acid * -hippuric acid * -homocysteine -Carboxymethyllysine (AGEs) -3-Carboxy-4-methyl-5-propyl-2-furan-propanoic acid -spermine

Abbreviations are; TNF: tumor necrosis factor; AGEs: advanced glycation end products. * refers to the molecules originating from colonic microbial metabolism.

**Table 2 toxins-12-00300-t002:** Impact of CKD, and PBURS on immune cell functions.

Cell Subset	CKD-Associated Phenotype	Impact of PBURS
p-Cresyl Sulfate	Indoxyl Sulfate
**Innate Immune Cells**
**Neutrophils**	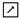 response to stimulation [53] 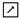 expression of TLR 2 and 4 [49,54] 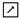 apoptosis [67] 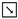 phagocytic functions [72,76,77]	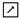 adhesion to endothelial cells and extravasation [66] 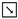 NADPH oxidase activity [73,74,75] 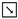 phagocytic functions [39]	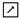 adhesion to endothelial cells and extravasation [66] 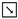 NADPH oxidase activity [73]
**Monocytes and macrophages**	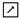 expression of TLR2 and 4 [49,54] 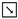 phagocytic functions [75,77]	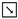 phagocytic functions [78,79]	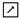 secretion of pro-inflammatory cytokines [57,58]
**Dendritic cells**	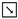 number [80,81,82] 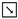 expression of costimulatory molecules [83,84] 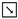 capacity to activate T cells [83,85]	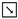 phagocytic function and presentation of antigen [78,79]	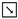 proliferationand expression of costimulatory molecules [86,87]
**Adaptive immune cells**
**Naïve T cells**	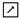 apoptosis [88] 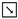 number [89,90,91] 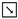 thymic output [90]	*Unknown*	*Unknown*
**Differentiated T cells**	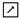 number of terminally differentiated [92,93] 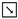 TCR repertoire diversity [94]	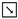 production of INFγ (Th1 cells) [95]	*Unknown*
**B cells**	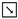 number of naïve and memory B cells [68,96,97] 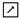 apoptosis [68,96,97] by decreased prosurvival signals [68,96,97]	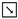 number of B cells [98]	*Unknown*

Abbreviations are; CKD: chronic kidney disease; PBURS: protein-bound uremic retention solutes; TLR: toll-like receptor; NADPH: nicotinamide adenine dinucleotide phosphate; TCR: T-cell receptor; INF: interferon; Th1: T helper phenotype 1; 
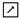
: increase; 
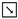
: decrease.

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
