# Peer review of "Chronic Kidney Disease-Associated Immune Dysfunctions: Impact of Protein-Bound Uremic Retention Solutes on Immune Cells"

_toxins, 2020, doi:10.3390/toxins12050300_

Round 1

Reviewer 1 Report

The reviewed manuscript pleasantly sums up the discussed issue. Below the comments:

Title
The title does not fully reflect the theme of the manuscript. It should be redrafted.

Introduction
Emphasize more the importance and legitimacy of the work.

Discussion
Not all abbreviations are developed at first use.
I propose to summarize the URS classification in a table.
Explain in more detail what are disturbances of tryptophan and tyrosine metabolism in CKD.
Finally, discuss further the relationship between inflammation and oxidative stress in CKD.
Chapters 4-6 should be summarized in the figure(s).
What is the clinical significance of the presented issue? Are URS used in the diagnosis of CKD? 

Reviewer 2 Report

This is an important topic given the unknown mechanism of immune dysfunction in CKD.

Comments:

Overall requires more debt, a through review of the topic

Clarify the definition pf microbiota, adaptive and innate immunity.

Table 1 lauches the paper and discussion and it is poorly organized, difficult to read, clarify columns and rows, there are also several abbreviations that are not defined in the text below the table. Requires complete reformatting

Must clarify what the definition of CKD is, what stage are you discussing?

Would delete the discussion of ESRD and dialysis (Line 100-106).  This paragraph is distracting from the main theme of the review.

What is meant in Line 127: "The uremic milieu alters the ability of endothelial cells to control the activation of coagulation?" This is not discussed in the reference sited.

Figure 1 is a summary and should be placed at the end with a discussion in the text. The references to Figure 1 in Section 3 does not fit.

Given the outbreak of Covid-19 and all the viruses patients are exposed to (Influenza, HIV, Hepatitis, RSV, ect.) recommend to expand this area and discuss in more detail the the cytokine levels in CKD.

The conclusion is that CKD is associated with defects in almost all components of the immune system. Recommend a section on therapy to improve immune dysfunction and future direction needed. 

Round 2

Reviewer 1 Report

The manuscript has been significantly improved.

I agree with the authors that uremic toxins are not routinely used in diagnostics, but the results of recent studies indicate their clinical usefulness, also in non-invasive diagnostics (e.g. in saliva). This topic should be briefly discussed, e.g. https://doi.org/10.1007/978-3-030-37681-9_10; 10.3390/toxins10010033.

Author Response

1/ The manuscript has been significantly improved.

We thank the reviewer for his positive appreciation of our efforts.

2/ I agree with the authors that uremic toxins are not routinely used in diagnostics, but the results of recent studies indicate their clinical usefulness, also in non-invasive diagnostics (e.g. in saliva). This topic should be briefly discussed, e.g. https://doi.org/10.1007/978-3-030-37681-9_10; 10.3390/toxins10010033.

In accordance with the reviewer’s suggestion, a new paragraph (and 3 new references) that covers the potential interest of non-invasive measurement of uremic toxins as biomarker has been added to the “perspective” section of the revised MS.

Change: lines 309-314

“Beyond their interest as therapeutic targets, PBURS could also be interesting biomarkers to stratify the risk of immune -mediated complications in CKD. Until recently, quantification of PBURS was not available in routine clinical practice due to its cost and the need of specific technologies (such as mass spectrometry [125]). However, the recent development of cheap assays that allow measuring non-invasively the levels of PBURS in the saliva, has the potential to rapidly change this situation [126,127]).“

Reviewer 2 Report

The revisions have improved the clarity of the manuscript.

I would accept for publication.

Author Response

The revisions have improved the clarity of the manuscript.

I would accept for publication.

We thank the reviewer for his constructive comments, which have helped improving the quality of the MS.